# Research on Combustion State System Diagnosis Based on Voiceprint Technology

**DOI:** 10.3390/s25103152

**Published:** 2025-05-16

**Authors:** Jidong Yan, Yuan Wang, Liansuo An, Guoqing Shen

**Affiliations:** School of Energy, Power and Mechanical Engineering, North China Electric Power University, No. 2 Beinong Road, Huilongguan, Changping, Beijing 102206, China; 120242102044@ncepu.edu.cn (J.Y.); 120222202113@ncepu.edu.cn (Y.W.); 120242102045@ncepu.edu.cn (L.A.)

**Keywords:** combustion, voiceprint, step index, CNN

## Abstract

This study investigates a multi-scenario combustion state diagnosis system based on acoustic feature extraction techniques. In this study, the voiceprint technology is applied to combustion condition monitoring for the first time, and an integrated approach for monitoring and diagnosis is proposed by combining multiple acoustic features, such as acoustic pattern features, step index P, and frequency-domain monitoring. In this study, a premixed hydrogen combustion test bed was built to simulate common combustion faults, and the corresponding acoustic features were collected and extracted. In this study, step index P and acoustic features are used for parallel diagnostic analysis, and CNN, ANN, and BP models are used to train the four states of flameout, flameback, thermoacoustic oscillation, and stable combustion, and the training diagnostic performance of each model is compared and analyzed using a confusion matrix. It is found that CNN has the strongest classification ability, can accurately distinguish the four states, has the lowest misclassification rate, has very strong generalization ability, and has a diagnostic accuracy of 93.49%. The classification accuracy of ANN is not as good as that of CNN, and there are local fluctuations during the training process. The BP neural network has a slower convergence speed and a high error rate in recognizing the flameback and thermoacoustic oscillations. In summary, the combustion state diagnosis system based on CNN model combined with acoustic features has optimal performance, and the combination of step index P and frequency-domain monitoring in the flameback diagnosis can improve the accuracy of combustion state identification and safety control level, which provides an important theoretical basis and practical reference in the field of combustion state diagnosis and is of profound significance to ensure the safe and efficient operation of the combustion process.

## 1. Introduction

The proposal of the dual-carbon target has introduced groundbreaking demands for the energy industry [1]. To adapt to the rapid growth and high variability of renewable energy sources in the grid, as well as significant fluctuations in electricity demand [2], thermal power units must not only fulfill their social responsibility for power generation and heat supply but also undertake the tasks of deep peaking. Under deep peaking, thermal-power-generating units have frequent load changes, and combustion faults such as flameback [3] and thermoacoustic oscillations [4,5] occur frequently [6,7,8]. These issues require higher standards for the safe operation of equipment and low ${NO}_{x}$ operation. In recent years, the rapid development of various new sensor types and diagnostic and analytical methods has provided the prerequisites for the comprehensive intelligent development of boilers. There is a growing demand for intelligent monitoring systems that can comprehensively, efficiently, and accurately assess the combustion status of boilers. Current flame monitoring systems can be categorized into three types based on the form of the sensor: optical radiation [9,10], thermodynamic [11], and electrical detection technologies [12]. Among these, using the acoustic features of combustion to monitor the flame status is a method based on the thermodynamic properties of combustion. Diagnosing the flame combustion status through its acoustic features offers the advantages of being non-contact, having a flexible sensor arrangement, being low-cost, not producing electromagnetic signals, and not interfering with the normal operation of the equipment [13,14].

In the process of flame combustion, sound waves are generated along with the fluctuation of gas and heat, and the sound wave spectrum is closely related to the combustion status of the flame. Research using acoustic signals to diagnose the combustion status, such as flameback and thermoacoustic oscillations, has a long history and encompasses a wide range of directions. In 1980, Detriche extracted and analyzed the sound of combustion, establishing the relationship between the flame acoustic spectra and burner diameters [15]; Yu measured the acoustic features of the flame at different locations using independent piezoelectric sensors, analyzing the changes in burning sound pressure level and spectral changes [16]; Singh found that the combustion acoustic signals were mainly concentrated in the high-temperature region by measuring the acoustic features at different locations and parameters such as time-averaged and fluctuating temperatures [17]; Ahmed et al. analyzed and monitored the acoustic signals of methane flames and heat release pulsations, exploring the coupling relationship between the acoustic features of flame combustion and the heat release pulsations by analyzing the signals in both the time and frequency domains [18]; Mondal used acoustic signals, performed dynamic analysis after Fourier transforming the signals, and introduced time and sound pressure sequences to classify the ignition features of the flame, using this parameter to classify the flame into stable and unstable status [19]; and Chandrachur proposed a method based on the time- and frequency-domain features of the sound for predicting and classifying identification of flame-quenching, stable status, and thermoacoustic oscillations. Past studies have primarily focused on analyzing the status change of combustion and the prediction of combustion accidents by studying the signal variation rules of combustion acoustic signals in the time domain, frequency domain, power spectrum, etc., with a fixed amount of air and the coupling of parameters such as heat release rate. However, the inability of the time-domain signals to highlight frequency variations and the frequency domain to show time variations, as well as the lack of diagnostic studies related to flameback acoustics due to the more hazardous nature of the flameback experiments, mean that the existing acoustic monitoring studies are unable to meet the requirements of combustion monitoring in power plants. Utilizing more accurate and efficient diagnostic methods for acoustic characterization has become the focus of research.

With further development of sound extraction algorithms and artificial intelligence, voiceprint technology has begun to be applied in various scenarios, including speech recognition [20], electric motors [21], gas turbine combustor [22], bear [23], aero-engines [24], gasoline engines [25], and equipment status monitoring, among others. In voiceprint monitoring, good data features can guarantee efficient and fast recognition analysis by the recognition system. There are many voiceprint features, including time domain, frequency domain, time–frequency domain, Mel-Frequency Cepstral Coefficients [26] (MFCCs), Gammatone Frequency Cepstral Coefficients (GFCC), and so on. Among them, MFCC contains both time–frequency-domain feature information and can be analyzed as one-dimensional data for recognition and prediction. Compared with image recognition, MFCC can further reduce the consumption of computational resources and computation time. In the field of status recognition, MFCC features have demonstrated good recognition rates and robustness [27]. At the same time, combined with the acoustic characteristics of the time-domain envelope response combustion state changes, the frequency-domain response to its changes in the distribution together constitute a combustion sound-pattern diagnostic system.

This study focuses on investigating multiple acoustic pattern trends under combustion faults such as flameout, thermoacoustic oscillations, and flameback, comparing the similarities, and arriving at an acoustic pattern diagnostic eye on method that can be applied to most combustion detections. Firstly, the combustion acoustic mode diagnosis data set is established through the preset combustion state in the laboratory, and the step index set Pi is calculated according to the data set. At the same time, the material damage is judged, and the acoustic characteristics such as time domain, frequency domain, and the MFCC of the specific fault diagnosis data set are calculated. Secondly, the centralized combustion fault diagnosis encountered in this study is tested using the collected test set to observe whether it is able to diagnose its state changes sensitively and accurately. This study attempts to use acoustic features in the field of speech recognition to monitor changes in combustion conditions and diagnose in real time whether to perform feed breaks to maintain stable and safe combustion and further deepens the combustion diagnostic model by using a variety of acoustic features.

## 2. Feature Extraction Algorithm

MFCCs are commonly used for the identification of speaker in voice recognition systems. Davis and Mermelstein launched them in the 1980s and they have been cutting-edge ever since [28,29,30]. MFCCs can simulate the nonlinear features of the human ear with a Mel filter bank and are usually used to extract signal features in sound signal processing [31]. It is the main parameter for voiceprint recognition at present. MFCC is primarily based on Mel filters for feature extraction; a Mel filter bank is a type of biomimetic filter constructed according to the features of the human ear’s auditory mechanism. As the human ear is sensitive to low-frequency sounds and less so to high-frequency sounds, the Mel filter bank features a denser arrangement at low frequencies and a sparser one at high frequencies. For the flame sound information, which primarily concentrates in the low-frequency range, MFCC offers better extraction accuracy and robustness.

### 2.1. Data Preprocessing

The audio signal is passed through a low-pass filter to attenuate the high-frequency noise signal, and the cutoff frequency is set to 5000 Hz. The sound signal is pre-emphasized to balance the high-frequency spectrum, making the signal spectrum flatter across the full domain. Pre-emphasis is applied to the audio signal to attenuate high frequency signals. The attenuation factor α is 0.97(1)y(n)=x(n)−αx(n−1), n=1,2,3…N, N=1024, 
x(n) is the signal corresponding to the nth sampling point of the original signal; y(n) is the signal corresponding to the nth sampling point after preprocessing; *N* is the number of points of the Fourier transform; and α is the attenuation coefficient.

### 2.2. MFCC Extraction Process

(1)The pre-emphasized signal in the previous step is divided into frames and windowed using Hamming window.


(2)
$$W(n)=0.54-0.46\cos\frac{2\pi n}{N-1},\;0\leq n\leq N-1,\;$$



(3)
y’(n)=y(n)×W(n), 


W(n): the value of the window function at the nth sampling point, the value of the y’(n): the preprocessed signal.

N: the number of sample points (i.e., window length) contained in each frame.

(2)The signal, now framed and windowed, is subjected to a Fast Fourier Transform, resulting in the frequency spectrum of the signal.


(4)
$$s(k)=\sum_{n=0}^{N-1}y’(n)e^{-j\frac{2\pi }{N}kn},\;0\leq k\leq N,\;$$


s(k): Fourier coefficients of the kth frequency point.

N: the number of transformed points.

(3)The energy spectrum is passed through a set of triangular filters designed according to the Mel scale. The frequency response of the triangular filters [$H_{m}(k)$] and the response formula for Mel frequency to actual frequency f are given by Equation (7). After band-pass filtering, the results are corrected using a logarithmic function to adjust for the non-linearity of sound intensity, thus obtaining the logarithmic energy output for each filter group. In this study, three types are designed, with M = 19, 39, 57, with f(m) corresponding to linear frequency, s(m) indicating the logarithmic energy output of each filter group.



(5)
$$H_{m}\left(k\right)=\left\{\begin{array}{l} 0,k≺f(m-1)\\ \frac{2(k-f(m-1))}{\left(f(m+1)-f(m-1))(f(m)-f(m-1)\right)},f(m-1)≺k≺f(m)\\ \frac{2(f(m+1)-k)}{\left(f(m+1)-f(m-1))(f(m)-f(m-1)\right)},f(m)≺k≺f(m+1)\\ 0,k≻f(m+1) \end{array}\right.,\;$$


(6)
$${\sum_{m=0}^{M-1}H}_{m}(k)=1,\;$$


(7)
$$Mel(f)=2595\times \log_{10}(1+\frac{f}{700}),\;$$


(8)
$$s(m)=\ln({\sum_{k=0}^{N-1}{\left|s(k)\right|}^{2}H}_{m}(k)),\;0\leq m\leq M.$$



(4)The cepstrum is computed through an Inverse Discrete Fourier Transformation, and the static MFCCs are derived from the logarithmic energy calculated in the previous step by applying a Discrete Cosine Transform (DCT). The formula for calculating static MFCC coefficients is as follows:

(9)$$\left\{\begin{array}{l} C(n)=\sum_{M=0}^{N-1}s(m)\cos\frac{n\pi (2m-1)}{2M}\\ n=1,2,\ldots ,L\\ m=1,2,\ldots ,M \end{array}\right.,\;$$
M is the number of filter banks; $H_{m}(k)$ is the frequency response of the mth triangular filter; f(m) is the corresponding linear frequency; and q(m) is the logarithmic energy output from each filter bank.

In this paper, three filter bank schemes are designed, which include 19, 39, and 57 banks to accommodate different frequency distributions, respectively.

(5)The calculation of the first-order derivative of MFCC is accomplished through differencing operations on the MFCCs. The formula for the first-order derivative of MFCC is as follows

(10)$$\delta_{c}\left[t\right]=\sum_{n=1}^{N}n•(c_{t+n}-c_{t-n}),\;$$
where ΔMFCC represents the first-order derivative of the Mel-Frequency Cepstral Coefficient (MFCC), N is the window size used for the differencing operation, and $\delta_{c}\left[t\right]$ denotes the time index.

### 2.3. The Step Index P

The envelope is often used in the analysis of vibration signals, in which the envelope obtained from the calculation of the original vibration signal in the high-frequency signal is filtered out better and can reflect the time-domain signal contour change rule with time. In this paper, the step index P is obtained by calculating the envelope index of its time-domain signal, and the change in its step index can accurately reflect the state transformation of its time domain. The specific process of its feature extraction is as follows:

(1)Obtain the original signal x(n) and perform the Hilbert transform on the original signal.

(11)$$H\{x(t)\}=\frac{1}{\pi }\cdot \mathrm{PV}\int_{-\infty }^{\infty }\frac{x(\tau )}{t-\tau }d\tau ,$$H{x(t)} is the Hilbert transform of the signal, PV denotes the principal value integral, and the singularity in the integral is taken into account.

(2)An imaginary part of the signal is obtained by the Hilbert transform, which is combined with the original signal to form an analyzed signal $\tilde{x}(t)$



(12)
$$\tilde{x}(t)=x(t)+j\cdot H\{x(t)\}.$$



(3)The step index P is the amplitude of the resolved signal


(13)
$$P(t)=|\tilde{x}(t)|=\sqrt{x{\left(t\right)}^{2}+H{\left\{x(t)\right\}}^{2}}.$$


## 3. Experimental

### 3.1. Experimental Apparatus

The fault combustion voiceprint test platform used in the research is shown in Figure 1, which consists of four parts. The first part is the material platform. The platform primarily includes air tank (providing a stable oxygen supply); hydrogen tank (main test gas); methane tank (responsible for igniting hydrogen flame and avoiding flameback); steam generator (inhibiting hydrogen combustion rate) and its storage tank; electronic gas flowmeter; and its control switch. The second part is the transportation pipeline platform, which is divided into two sides. One side is the gas pipeline input to the gas flowmeter, which is gradually adjusted to the required ratio and working condition through the gas flowmeter. The other side is the premixer, which premixes the gas in advance and fully premixes the gas to make it fully burn, wherein the burner is a 4 by 4 array burner with a burner tube diameter of 2 cm and a spacing of 5 cm between tube diameters. The gas fuel flow is measured by a flowmeter with an accuracy of ±1%, and its range is 0–300 SLM. Here, SLM is used to represent the flow rate per minute in standard conditions (0 °C, 1 atm). The accuracy of the device for measuring air is ±1%, and the range is 0–300 SLM gas flowmeter. SLM is a unit used to express liters per minute at standard conditions (0 °C, 1 atm). The third part is the combustion chamber, which contains an array burner in the middle. The last part is the voiceprint test platform, including the microphone acquisition device (using rode’s microphone, the model is RODE NT g4+, which can better suppress external noise), camera, data acquisition card (DAQ), computer, and a connecting cable. In order to avoid overheating the microphone, the microphone is 50 cm away from the flame center. Also, in order to ensure the accuracy and reliability of the experiment, the location of all the equipment was kept as unchanged as possible. There is an observation port on the side, which is convenient for observing the combustion state. The ventilation pipe is connected to the flue mouth to exhaust smoke in time. In this study, all equipment except microphones and computers were customized and produced in Beijing, China.

### 3.2. Experimental Design

The position of the flame combustion center fluctuates the most. The microphone is used to align the flame center position, and the sampling frequency is 44,000 Hz. The heat release rate of hydrogen combustion is much larger than the gas transport rate, which is more prone to combustion failures, such as flameback. Methane is supplied in the early stage of ignition to ensure stable combustion of the flame, and hydrogen supply is gradually increased. However, for safety testing, saturated water vapor is required to reduce the proportion of hydrogen and its reaction rate. The design and practice of this experiment are mainly to test two common combustion faults of flame flameback and thermoacoustic oscillation. By maintaining the stable operation of methane, air, and water vapor, the proportion of hydrogen is gradually increased, and the occurrence of hydrogen flameback and thermoacoustic oscillation and the change of voiceprint state under different ratios are tested. In order to ensure the reliability and safety of the sample, each group of working conditions was tested in two categories. Under the condition of keeping the flow rate of hydrogen and air constant, the flow rate of water vapor was changed to make it easier to temper and to verify the effect of two different water vapor flow rates on flameback production. At the same time, the emergency brake switch was installed under the gas flow controller. The hydrogen supply switch was turned off within two seconds of flameback, the air and water vapor were continuously supplied, and the residual gas in the pipeline and premixer was purged. This experiment is divided into five groups of working conditions, which represent their different ratios. The sampling classification is shown in Table 1, and the corresponding air coefficient is calculated. Therefore, using the mass flow rate of air and fuel ($M_{A}$ and $M_{F}$, respectively) to calculate the value of the air coefficient, $\phi ={\left(\frac{M_{A}}{M_{F}}\right)}_{reality}/{\left(\frac{M_{A}}{M_{F}}\right)}_{theoretical}$. The fuel used in this study consisted of hydrogen at 99% purity and methane at 99% purity.

In this experiment, a total of 10 working conditions were designed, each of which can be divided into two categories. These two types keep the air coefficient constant and change the amount of water vapor in the pipe at the same time, which makes the experiment easier to flameback. In this study, water vapor mainly inhibits hydrogen combustion. Under the same ratio, in order to make the combustion more prone to flameback and thermoacoustic oscillation, the water vapor content is reduced to observe whether combustion failure becomes more likely. At the same time, water vapor also provides a safety mechanism for the occurrence of combustion faults. When flameback occurs, continuous supply of water vapor can further purge the gas supply pipeline, reduce the temperature of the burner and the experimental pipeline, and reduce the risk of deflagration of the burner.

### 3.3. Flow Chart of Combustion Voiceprint Intelligent Diagnosis System

The voiceprint intelligent diagnosis system studied in this paper is mainly divided into two steps. The detailed research route is shown in Figure 2. The first step is the laboratory combustion fault voiceprint data acquisition research. Through the laboratory preset combustion fault sound signal, the amplitude step index P and the corresponding voiceprint features are calculated and extracted. In this experiment, there are three states: pre-set and experimental, flameout state (distinguished from the state of non-ignition, which is transformed from combustion state or combustion fault state), and the thermoacoustic oscillation and flameback state. After the acquisition, the corresponding indexes are calculated respectively to form a step data set. By calculating the step index, it provides a data basis for the intelligent diagnosis system to judge whether to disconnect the combustion. At the same time, the sound signal of the fault state is preprocessed by noise reduction, and the frequency domain, time–frequency domain, and MFCC voiceprint features are extracted and constitute the data set of combustion fault voiceprint diagnosis as a whole. The second step is to run the combustion state diagnosis system in real time. The acoustic signal of combustion fault preset by the laboratory is used as the diagnostic training set, and the real-time acquisition signal is used as the test signal. The size range and various characteristics of P are judged separately, and multiple signal characteristics are fused to obtain the final combustion state and determine whether it is necessary to alarm.

### 3.4. Input Parameters of Combustion Voiceprint

The typical working condition 0.90–1 is selected as the main body of data analysis. In order to reflect the normal combustion state, the data within 10 s before and after are selected as the target data for the analysis. In order to obtain more accurate data, the microphone used in this experiment is a gun microphone, which has the characteristics of unidirectional acquisition and high frequency filtering. In order to unify the same conclusion under different ratios, all test data need to be normalized.

#### 3.4.1. Amplitude Step Index P

The amplitude step index P, as the most important basis for fault diagnosis in this study, has the following characteristics. As shown in Figure 3, when thermoacoustic oscillation occurs during combustion, it has the following characteristics: First, when a combustion fault occurs, the amplitude of the time-domain signal reaches the limit of the device test; during normal combustion, the time-domain signal will remain within the device test limit. The second envelope value remains between 0 and 1 before the thermoacoustic oscillation occurs; when the thermoacoustic oscillation occurs, some signals will break through the limit of 1, but the duration is less than 0.1 s. When the thermoacoustic oscillation occurs, the envelope value will remain within the range of 0.3–2 and last for more than 0.1 s. As shown in Figure 3 and Figure 4, when flameback occurs during the combustion process, it also has the following characteristics: the first time-domain signal reaches the limit value of the equipment test, and the normal combustion is maintained within the equipment test limit, similar to the thermoacoustic oscillation time-domain signal. When the second envelope value is not tempered, its state is more stable than the thermoacoustic oscillation, and its value remains between 0 and 1; during the combustion flameback, the maximum value of the envelope value abruptly changes to more than 2, and the change in the value is maintained between 0 and 2.

Through the above analysis, it is found that the difference between flameback and thermoacoustic oscillation lies in whether there is an unstable state and the change range of envelope value on the eve of fault occurrence. The change in flameback envelope value is between 0 and 2, while the thermoacoustic oscillation is between 0.3 and 2.

#### 3.4.2. Frequency-Domain Signal

Monitoring the change in the dominant frequency in the combustion frequency domain has been the mainstream monitoring method in recent years. For stable combustion, shown inn Figure 5, the signal is a broadband signal with multiple dominant frequency distributions. When the combustion state changes abruptly, the frequency-domain distribution becomes a single main frequency, and the main frequency becomes 238 Hz in the flameback state, as shown in Figure 6, and the main frequency becomes 221 Hz in the thermoacoustic oscillation state, as shown in Figure 7. By monitoring the change in its main frequency, the change in combustion state can be better judged. However, in experiments, with the change in the ratio and the different air intake of water vapor, it also causes a certain deviation of the thermoacoustic oscillation and flameback main frequency generated by different ratios, and the distribution of other sub-frequencies is different. The frequency-domain signal can determine whether the combustion is stable or unstable, but it cannot determine what kind of instability has occurred.

#### 3.4.3. MFCC Feature

In this study, three common faults of combustion diagnosis—thermoacoustic oscillation, flameback, and flameout—were designed and studied. In the above research, the time-domain characteristic step index P can determine whether the fuel supply needs to be disconnected to avoid deflagration failure. The frequency distribution can better judge the difference between thermoacoustic oscillation, flameback, and normal combustion. However, only relying on the frequency distribution as the input signal not only increases the amount of diagnostic data, but also increases the recognition time. Under various ratios, the frequency distribution is similar but inconsistent, which increases the difficulty of recognition and reduces the accuracy of the model. In order to improve the performance of the model, the MFCC feature, as an acoustic feature that not only has time characteristics, but also can see its frequency distribution, has become the main force of model training.

Figure 8, Figure 9 and Figure 10 show the original data diagrams of thermoacoustic oscillation, flameback, and normal combustion, respectively. From these diagrams, it can be observed that the patterns differ under various states—particularly between thermoacoustic oscillation and flameback. Notable differences appear in MFCC frames 20–25 and 28–30, where the distribution deviates significantly from that of flameback. The MFCC feature distribution of stable combustion is clearly distinct from both thermoacoustic oscillation and flameback, with the voiceprint MFCC of stable combustion displaying a broadband pattern.

## 4. Results and Discussion

In order to build a more practical combustion monitoring system, this paper uses the step-order data set Pi for the diagnosis of combustion state. The other features are used as the training data set of the model for training and classification diagnosis in which the labels of all states are manually labeled, and the predicted values obtained by feeding the model training are the states diagnosed by the system.

### 4.1. The Training Performance of Different Models Under Different Working Conditions

In this study, three commonly used models, Backpropagation Neural Network (BP), Artificial Neural Network (ANN), and Convolutional Neural Network (CNN), were adopted. BP neural network is a multi-layer perceptron (MLP) based on gradient descent optimization algorithm, which uses error back propagation algorithm (BP algorithm) for training and model diagnosis. BP network is widely used in non-linear regression and classification tasks, but it easily falls into local optimum during model training, and the training time is long. At the same time, BP is sensitive to the selection of hyperparameters. The ANN model is one of the most basic neural network structures. It imitates the working mode of human brain neurons for information transmission and data processing. The ANN model is suitable for various data modeling tasks, but its learning ability is limited by the depth of the network. It struggles to handle high-dimensional complex data and faces challenges in training large-scale models for big data. Convolutional Neural Networks (CNNs) are a class of deep learning models particularly well suited for processing image and time-series data. They are capable of efficiently extracting local features and exhibit strong performance in training on both one-dimensional sequences and graph-structured data. In this study, various models were employed to train the experimental data sets, and a comparative analysis of their performance is presented in Figure 11 and Table 2.

From the training trend of the model, the convergence speed of the CNN model is relatively fast, the loss function can be stabilized in fewer training rounds, and its feature extraction ability is strong. The ANN model shows local fluctuations during the training process, indicating that there may be local optimal problems in the optimization process, and the single model can still converge well. The training convergence speed of BP neural network model is slow, and the loss decreases slowly during the training process. It may take longer training time to achieve the ideal accurate diagnosis effect.

### 4.2. Diagnosis Results of Different Models

In this study, three models of CNN, BP neural network, and ANN were used to classify and diagnose the four states of flameout, flameback, thermoacoustic oscillation, and stable combustion, and the performance of each model was compared and analyzed by confusion matrix. On the whole, the confusion matrix of the CNN shows that its classification ability is the best, the recognition rate is the highest on flameback and thermoacoustic oscillation, and it has excellent feature extraction and diagnosis ability. Both ANN and BP have a certain degree of misjudgment in the classification of flameback and thermoacoustic oscillations. Among them, BP has the highest number of misjudgments of thermoacoustic oscillations, indicating that its ability to distinguish complex categories of diagnosis is relatively weak. The final results show that CNN has the best diagnostic performance, and its training convergence speed is fast and stable. The CNN model can accurately distinguish between various combustion states, achieving a diagnostic accuracy as high as 93.49%. In comparison, although the BP neural network possesses some diagnostic capability, it suffers from a slow convergence speed, significant training fluctuations, and a high misclassification rate between flameback and thermoacoustic oscillation. The ANN model performs slightly better than BP in classifying thermoacoustic oscillation, but its overall generalization ability remains weak. All models are able to correctly identify flameout and stable combustion states; however, distinguishing between flameback and thermoacoustic oscillation remains the main challenge. A comprehensive analysis indicates that the CNN model demonstrates the best applicability for this monitoring task. The advantages and disadvantages of model diagnosis are shown in Figure 12 and Table 3.

## 5. Summary, Conclusions, and Future Work

In this study, the performance of CNN, ANN, and BP models in the combustion state diagnosis system was compared and analyzed, and their applicability was discussed in combination with the training trend. At the same time, a method for comprehensive utilization of voiceprint characteristics, step index P and frequency-domain monitoring is proposed to improve the accuracy of combustion state diagnosis. The main conclusions are as follows:(1)The step index P exhibits high specificity in the flameback state, allowing it to effectively distinguish abnormal changes in the combustion state. It serves as a decision-making tool to determine whether to disconnect the fuel supply, thereby enhancing combustion safety.(2)By monitoring the frequency-domain characteristics of the signal, the change in the burner state can be accurately identified. Compared to time-domain analysis, frequency-domain information provides a more intuitive reflection of the dynamic changes in combustion state, offering valuable insight for combustion stability analysis.(3)In the combustion state diagnosis system, CNN demonstrates the best classification ability, effectively distinguishing between the four states: flameout, flameback, thermoacoustic oscillation, and stable combustion. Its confusion matrix reveals the lowest misjudgment rate and the strongest generalization ability. While ANN performs slightly worse than CNN in classification accuracy, it exhibits some local fluctuations during the training process, potentially influenced by local optimization. The BP neural network, on the other hand, has a slow training convergence speed, a high misjudgment rate for flameback and thermoacoustic oscillation states, and relatively weak overall diagnostic performance. In summary, this study shows that the combustion state diagnosis system based on the CNN model combined with voiceprint features has the best performance, and the combination of step index P for flameback diagnosis and frequency-domain monitoring can further enhance the accurate identification and safety control of combustion state.

This study investigates the feasibility of applying acoustic pattern techniques to monitor the combustion process under laboratory conditions. The results show that acoustic features can effectively reflect the combustion state and provide a potential new method for real-time diagnosis of the combustion process. The data processing in this study was based on the experiments after which the feasibility of its acoustic pattern technique applied to practical diagnosis was explored. However, practical combustion monitoring needs to cope with challenges such as background noise interference and ensure that relevant acoustic features can be robustly extracted. In addition, there is a need to further establish clear correlations between acoustic features and combustion temperature control and product composition.

## Figures and Tables

**Figure 1 sensors-25-03152-f001:**
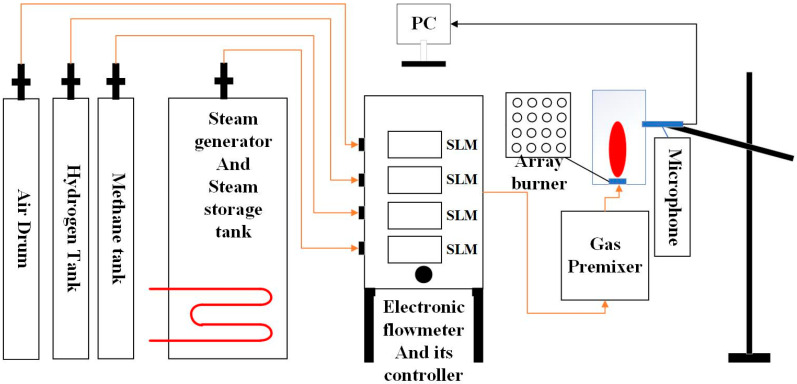
Experimental diagram of combustion diagnosis system.

**Figure 2 sensors-25-03152-f002:**
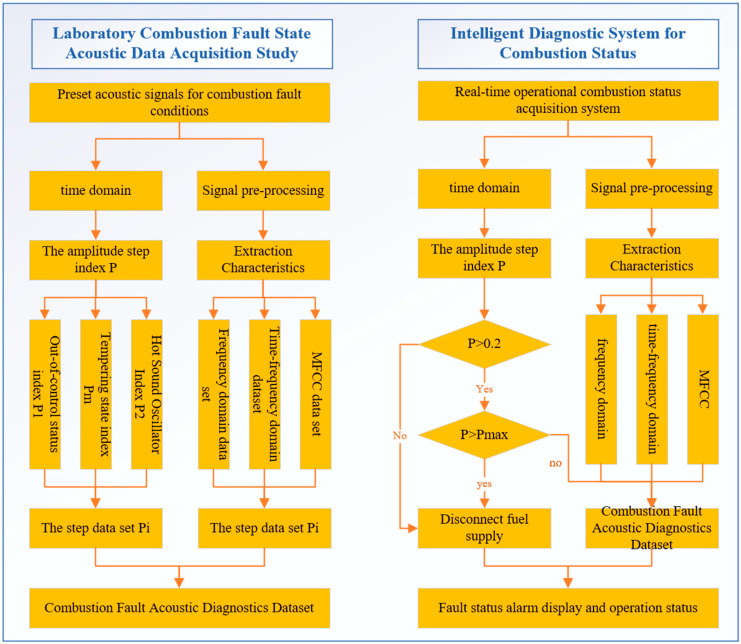
Research flow chart of combustion fault diagnosis.

**Figure 3 sensors-25-03152-f003:**
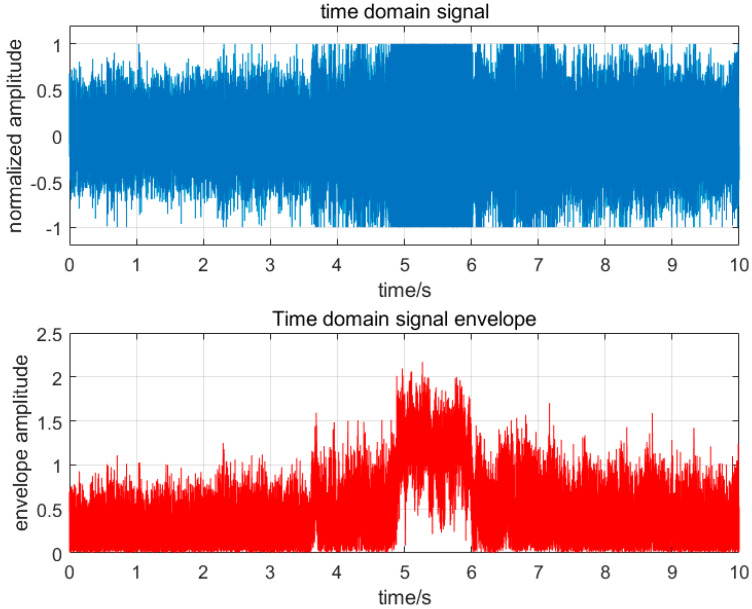
Thermoacoustic oscillation time−domain and envelope diagram.

**Figure 4 sensors-25-03152-f004:**
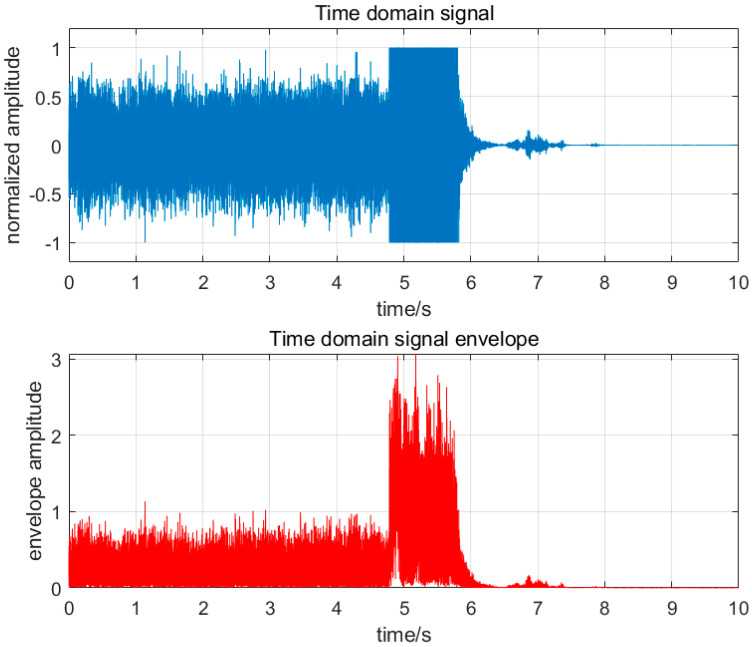
Flameback time−domain and envelope diagram.

**Figure 5 sensors-25-03152-f005:**
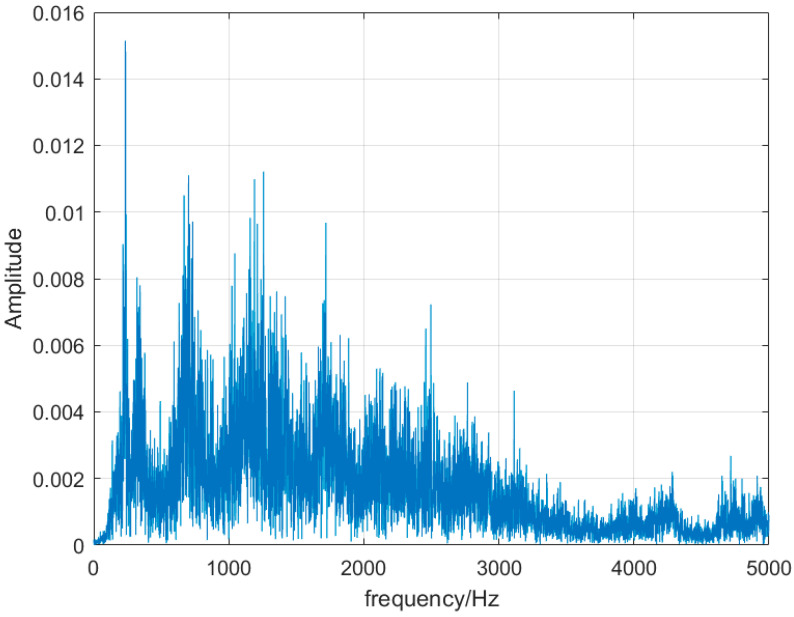
Stable combustion schematic diagram.

**Figure 6 sensors-25-03152-f006:**
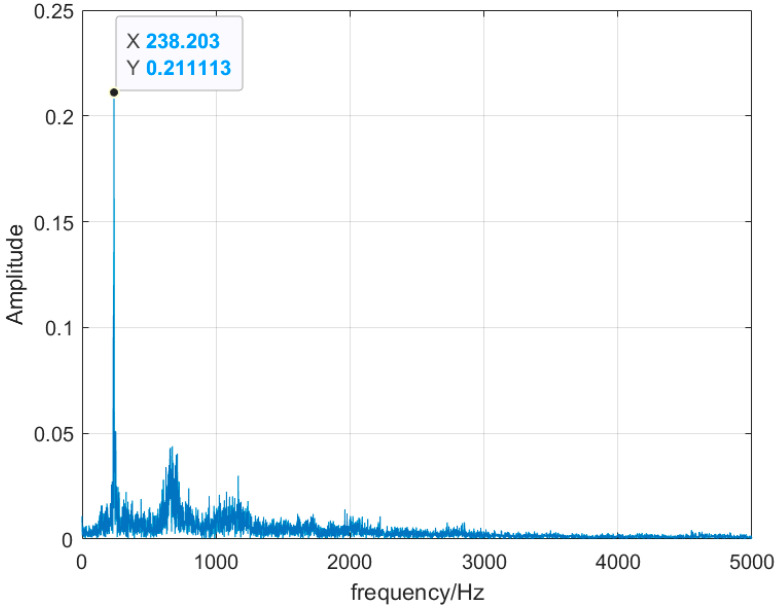
Flameback state frequency-domain diagram.

**Figure 7 sensors-25-03152-f007:**
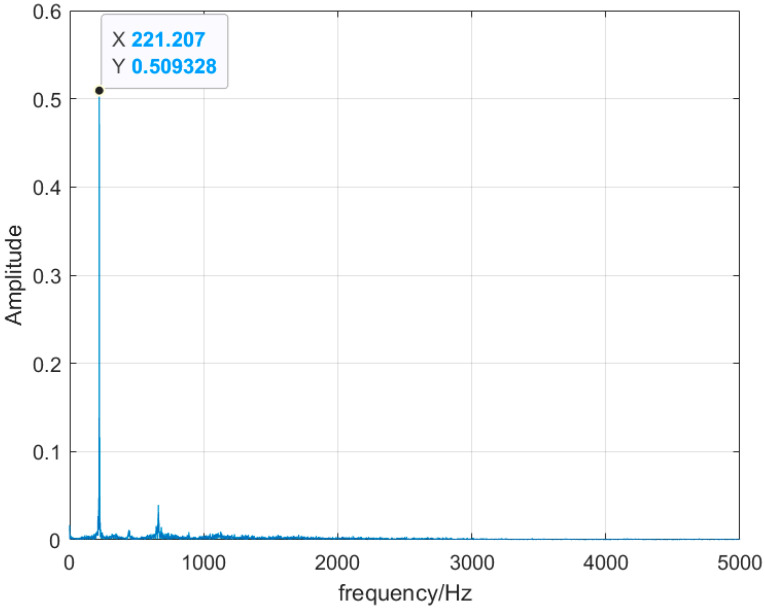
Thermoacoustic oscillation frequency-domain diagram.

**Figure 8 sensors-25-03152-f008:**
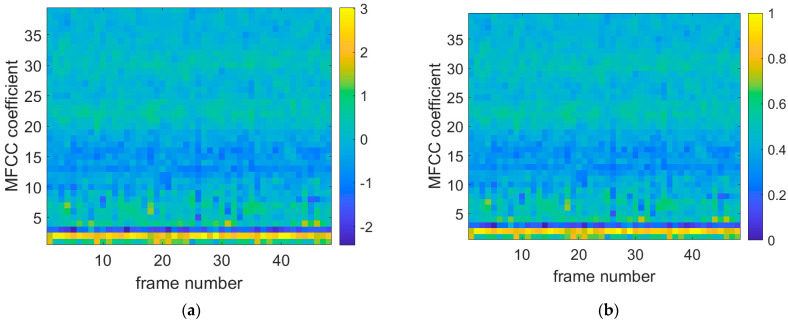
Combustion oscillation MFCC data diagram. (**a**) Thermoacoustic oscillation original MFCC data diagram; (**b**) thermoacoustic oscillation normalized MFCC data plot.

**Figure 9 sensors-25-03152-f009:**
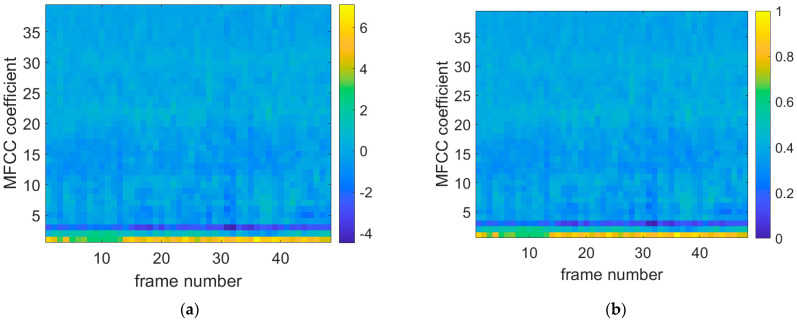
Flameback MFCC data diagram. (**a**) The original MFCC data diagram of flameback state; (**b**) flameback normalized MFCC data graph.

**Figure 10 sensors-25-03152-f010:**
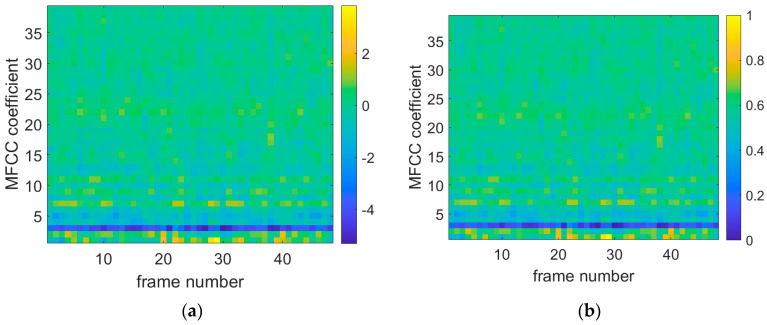
Normal combustion MFCC data diagram. (**a**) The original MFCC data of stable combustion; (**b**) stable combustion normalized MFCC data map.

**Figure 11 sensors-25-03152-f011:**
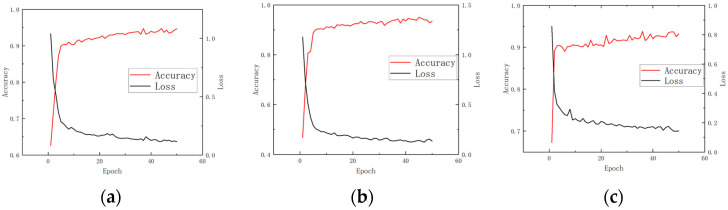
Three models of neural network training performance trend chart. (**a**) ANN neural training performance chart; (**b**) BP neural training performance chart; (**c**) CNN neural training performance chart.

**Figure 12 sensors-25-03152-f012:**
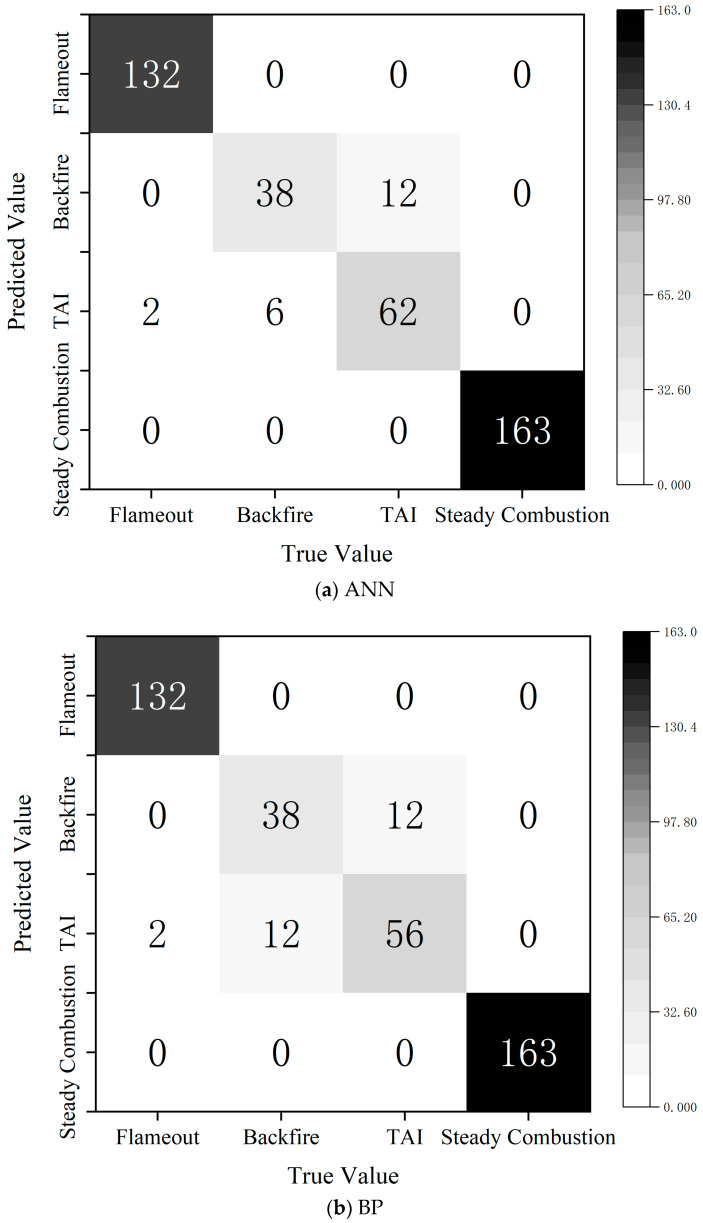
Different models’ monitoring confusion matrix diagrams.

**Table 1 sensors-25-03152-t001:** Fault combustion experiment condition design table.

Condition	$H_{2}$ (SLM)	$O_{2}$ (SLM)	$H_{2}O-A$ (SLM)	$H_{2}O-B$ (SLM)	Air (SLM)
0.95–1	112	59	30.4	30.4	171
0.95–2	112	59	22.4	30.4	171
0.90–1	112	62	30.4	30.4	173
0.90–2	112	62	21.4	30.4	173
0.85–1	112	65	30.4	30.4	176
0.85–2	112	65	20.4	30.4	176
0.80–1	112	69	30.4	30.4	180
0.80–2	112	69	19.4	30.4	180
0.75–1	112	74	30.4	30.4	185
0.75–2	112	74	18.4	30.4	185

**Table 2 sensors-25-03152-t002:** Comparison table of training performance of different models.

Model	Final Accuracy Rate	Loss of Convergence Speed	Stability	Generalization Ability
ANN	About 90%	The slowest convergence.	Has a rather great fluctuation.	Poor, lack of generalization ability.
BP	91–93%	The convergence is slow, and the first 10 rounds fluctuate greatly.	Medium, but the loss fluctuates greatly.	Good, but easy to fall into local optimum.
CNN	93.49%	The convergence is the fastest, and it is mostly stable after 10 rounds.	The most stable, no obvious shock.	Optimal, training error and test error minimum.

**Table 3 sensors-25-03152-t003:** Misclassification summary of three models.

Misclassification Mode	The Misclassification Number of ANN Model	The Misclassification Number of BP Model	The Misclassification Number of CNN Model	Analysis
Misclassification of misfire extinguishment	0	0	0	All models can accurately classify the flameout state.
Flameback is mistaken for thermoacoustic oscillation	12	12	6	CNN is the least, BP and ANN are more, indicating that BP/ANN has a weak ability to classify flameback.
Thermoacoustic oscillation is misjudged as flameback	6	12	9	CNN has fewer misjudgments, and ANN performs slightly better than BP in this category.
Thermoacoustic oscillation is mistaken for flameout	2	2	0	CNN did not have this problem, and BP and ANN made misjudgments.
Misclassification of stable combustion	0	0	0	All models can accurately classify stable combustion states.

## Data Availability

Data will be made available on request.

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
