# Peer review of "Research on Combustion State System Diagnosis Based on Voiceprint Technology"

_sensors, 2025, doi:10.3390/s25103152_

Round 1
Reviewer 1 Report
Comments and Suggestions for Authors
Thank you for this submission. While it presents some interesting results, this manuscript has several major flaws that currently preclude publishing it. Most importantly, I was unable to understand much of the text. Here are a few examples of phrases that I could not make sense of:
- Line 28: the proposed system is said to be able to "break the material in time"
- Line 219: gas flow rates were "verified each other"
- Line 229: "the fuel used in this study is 99% hydrogen and 99% methane"
- Line 231: "In the case of keeping the ratio of oxygen and oxygen unchanged ..."
- Line 273: "First, the time domain signal reaches the limit value of the equipment test"
I should also note that Line 408 includes a statement crediting an automated translation service.
I suggest a thorough review of this paper by an editor to ensure that your meaning is understood and that the text is free of errors.
Neither your title nor your abstract specifies what kind of combustion system you are investigating. This becomes clear later, but I recommend describing the system (a premixed hydrogen burner) and its applications (power generation) in the title and/or abstract.
You refer to "tempering" as an undesirable state for the combustion system. Tempering is a method of strengthening metals through heat treating, and I do not understand what you mean by it here. Are you referring to quenching of the flame, or lean blow-out?
If I understand correctly, this is the first attempt to apply voiceprint technology to combustion systems for energy production. If that is correct, you should emphasize that fact in the introduction. You mention that voiceprint technology has been applied to several other combustion experiments, but you don't elaborate on how it has succeeded or failed in those efforts. How do the previous studies inform what you are doing now?
The data processing section is very difficult to understand, especially for a reader who is not an acoustic scientist. I recommend starting at the beginning and carefully defining each term that you use. For instance, in Equation (1), is x(n) a time-domain signal? Why are there 1,024 time steps?
The experimental setup is not described in sufficient detail so that a reader could replicate your experiment and confirm your results. For example, readers will likely want to know what specific microphone you used and exactly where it was placed relative to the burner. Other important but missing pieces of information include the size and spacing of tubes in the array burner. Combustion state images are mentioned but never shown.
These issues in the setup description make it very difficult to assess the validity and impact of the results. Also, it was unclear to me in the end whether the voiceprint system had been implemented as a real-time alarm system, or whether all data processing occurred after the fact.
Author Response
Comments 1: While it presents some interesting results, this manuscript has several major flaws that currently preclude publishing it. Most importantly, I was unable to understand much of the text. Here are a few examples of phrases that I could not make sense of:
- Line 28: the proposed system is said to be able to "break the material in time"
- Line 219: gas flow rates were "verified each other"
- Line 229: "the fuel used in this study is 99% hydrogen and 99% methane"
- Line 231: "In the case of keeping the ratio of oxygen and oxygen unchanged ..."
- Line 273: "First, the time domain signal reaches the limit value of the equipment test"
I should also note that Line 408 includes a statement crediting an automated translation service.I suggest a thorough review of this paper by an editor to ensure that your meaning is understood and that the text is free of errors.
Response 1: Thank you for your suggestions. According to your comments, I have made the following amendments to the above questions: in line 28, remove “break the material in time” to avoid ambiguity; in line 219, remove “verified each other” and change it to a more appropriate description, the amendment is now in line 235; in line 229, the description of “ the fuel used in this study is 99% hydrogen and 99% methane” is inaccurate. the fuel used in this study is 99% hydrogen and 99% methane” in line 229 was inaccurate, so the description was changed to make it more appropriate, and the change is now in line 243; to the original line 231, ‘In the case of keeping the ratio of oxygen and oxygen unchanged ...’, the description was changed to make it more appropriate. unchanged ...” Change the original line 231 “In the case of keeping the ratio of oxygen and oxygen unchanged ...” to a more readable and fluent statement, the revised statement is in line 245; the original line 273 “First, the time domain signal reaches the limit value of the equipment test” is changed to a more appropriate description, the changed content is in line 288. And the automatic translation part of this paper has been further revised and embellished.
Comments 2: Neither your title nor your abstract specifies what kind of combustion system you are investigating. This becomes clear later, but I recommend describing the system (a premixed hydrogen burner) and its applications (power generation) in the title and/or abstract.
Response 2: Thank you for your suggestions. Based on your comments I have added a specific description and its resultant application in the abstract.
Comments 3: You refer to "tempering" as an undesirable state for the combustion system. Tempering is a method of strengthening metals through heat treating, and I do not understand what you mean by it here. Are you referring to quenching of the flame, or lean blow-out?
Response 3: Thank you for your suggestions. As per your suggestion, I have changed “Tempering” to “flameback” in the original article and replaced it in its entirety, which is a common term used in combustion research, and added appropriate literature support in the introduction.
Comments 4: If I understand correctly, this is the first attempt to apply voiceprint technology to combustion systems for energy production. If that is correct, you should emphasize that fact in the introduction. You mention that voiceprint technology has been applied to several other combustion experiments, but you don't elaborate on how it has succeeded or failed in those efforts. How do the previous studies inform what you are doing now?
Response 4: Thank you for your suggestion . According to your suggestion, I have added the corresponding descriptions in the Introduction and Abstract, and analyzed the monitoring accuracy of its different models for different combustion states in detail in 4.2. Diagnosis Results of Different Models. And at the end 5. Summary, Conclusions, and Future Work explains the future outlook of this experiment.
Comments 5: The data processing section is very difficult to understand, especially for a reader who is not an acoustic scientist. I recommend starting at the beginning and carefully defining each term that you use. For instance, in Equation (1), is x(n) a time-domain signal? Why are there 1,024 time steps?
Response 5: Thank you for your suggestion, according to your suggestion I have made a comprehensive arrangement of the algorithm principle in the chapter 2. Feature Extraction Algorithm, explaining all the variables that appear to make the logic smoother.
Comments 6: The experimental setup is not described in sufficient detail so that a reader could replicate your experiment and confirm your results. For example, readers will likely want to know what specific microphone you used and exactly where it was placed relative to the burner. Other important but missing pieces of information include the size and spacing of tubes in the array burner. Combustion state images are mentioned but never shown.These issues in the setup description make it very difficult to assess the validity and impact of the results. Also, it was unclear to me in the end whether the voiceprint system had been implemented as a real-time alarm system, or whether all data processing occurred after the fact.
Response 6: Thank you for your suggestion. Based on your suggestion, I have added a more detailed description of the equipment in 3.1. Experimental Apparatus, and a detailed explanation of the equipment tested. In 5. Summary, Conclusions, and Future Work, I have added a description that all data processing for this experiment took place after the experiment, and that the corresponding monitoring system code is currently being developed based on the results of the experiment.
Reviewer 2 Report
Comments and Suggestions for Authors
The article deals with the issues of acoustic monitoring of combustion processes in boilers at power plants. The main direction of research is related to the use of more accurate and effective diagnostic methods to determine acoustic characteristics has become the main direction of research. Artificial intelligence is used to process acoustic information. The following negative phenomena affecting the acoustic picture of the combustion process are considered: flashover, thermoacoustic oscillations and backfire.
It should be noted that the proposed approach is only one of the approaches. The real problem of the combustion process to be solved is more complex. It is necessary to take into account two other factors that allow more complex control of the combustion process to optimize the combustion process. First of all, it is necessary to control the temperature in different places of the boiler. It is also necessary to control the end product of the combustion process. Then all three control possibilities will allow to optimize the combustion process in boilers of different power or heat generation devices. Therefore, there is a proposal to the authors to revise the experimental part of the paper taking into account this comment.
It should also be noted a large number of technical errors in the design of the article due to inattention of the authors. It is recommended to bring it in compliance with the rules of article design:
(1) line 35: the energy industry [1]. To adapt to the rapid growth and high variability of renewable en-
(2) line 36: ergy sources in the grid, as well as significant fluctuations in electricity demand [2], thermal
(3) line 39: operational status changes frequently, and faults such as Lean-blow out (LBO) [3-5], and ther-
(4) line 40: moacoustic instability [6-8] (TAI) occur frequently. These issues require higher standards for
(5) line 47: ation [9, 10], thermodynamic [11],and electrical detection technologies [12]. Among these, using the
(6) line 52: with the normal operation of the equipment [13, 14].
(7) line 56: such as LBO and TAI, has a long history and encompasses a wide range of directions. In
(8) line 58: ship between the flame acoustic spectra and burner diameters [15]; Yu measured the acoustic
(9) line 60: lyzing the changes in burning sound pressure level and spectral changes [16]; Singh found
(10) line 63: time-averaged and fluctuating temperatures [17]; Ahmed et al. analyzed and monitored the
(11) line 66: sations by analyzing the signals in both the time and frequency domains [18]; Mondal used
(12) line 69: flame, using this parameter to classify the flame into stable and unstable status [19]; Chan-
(13) line 72: thermoacoustic oscillations [13]. Past studies have primarily focused on analyzing the status
(14) line 84: recognition [20], electric motors [21], gas turbine combustor [22], bear [23], aero-engne [23], gasoline en-
(15) line 85: gines [25], and equipment status monitoring among others. In voiceprint monitoring, good
(16) line 88: frequency domain, Mel-Frequency Cepstral Coefficients [26] (MFCC), Gammatone Fre-
(17) line 93: recognition, MFCC features have demonstrated good recognition rates and robustness [27].
(18) line 114: since [28-30]. MFCC can simulate the nonlinear features of the human ear with a Mel filter
(19) line 115: bank and is usually used to extract signal features in sound signal processing [31]. It is the
(20) line 157: start of line without indentation
(21) line 163: where ΔMFCC is the first-order derivative of MFCC, N is the window size for the differ-
(22) line 175: start of line without indentation
(23) line 186: includes air tank (providing a stable oxygen supply), hydrogen tank (main test gas),
(24) line 187: methane tank (responsible for igniting hydrogen flame and avoiding tempering), steam
(25) line 188: generator (inhibiting hydrogen combustion rate) and its storage tank, electronic gas flow-
(26) line 196: per minute at standard conditions (0 °C, 1 atm). The third part is the combustion chamber,
(27) line 194: an accuracy of ±1%, and its range is 0-300 SLM; the accuracy of the device for measuring
(28) line 195: air is ±1%, and the range is 0-300SLM gas flowmeter. SLM is a unit used to express liters
(29) line 226: ard conditions (0 °C, 1 atm). Therefore, using the mass flow rate of air and fuel (respec-
(30) line 227: tively MA and MF) to calculate the value of the air coefficient,
(31) line 228: the fuel used in this study is 99% hydrogen and 99%
(32) line 272: study, has the following characteristics. As shown in Figure 3, when thermoacoustic os-
(33) line 299: such as Figs .5, the signal is a broadband signal with multiple dominant frequency distri-
(34) line 302: ing state, as shown in Figs. 6, and the main frequency becomes 221 Hz in the thermoacous-
(35) line 303: tic oscillation state, as shown in Figs. 7 bBy monitoring the change of its main frequency,
(36) line 312: Figure 6. Empering state frequency domain diagram.
(37) line 314: Figure 7. Hermoacoustic oscillation frequency domain diagram.
(38) line 328: In Figs. 8 ,Figs. 9, Figs. 10, the original data diagrams of thermoacoustic oscillation,
(39) line 356: vantages of model training are shown in Fig. 11 and Table 2.
(40) line 380: various states, and the diagnostic accuracy is as high as 93.49%. In contrast, although BP
(41) line 399: (1) The step index P shows high specificity in the tempering state, which can effec-
(42) line 402: safety. Through the monitoring of the frequency domain characteristics of the signal,
(43) line 404: (2) By monitoring the frequency domain characteristics of the signal, the change of
(44) line 408: sis.
(45) In Figures 8-10, the font size on the axes needs to be increased.
Comments on the Quality of English Language
The English could be improved (strongly).
Author Response
1.Point-by-point response to Comments and Suggestions for Authors
Comments 1: The real problem of the combustion process to be solved is more complex. It is necessary to take into account two other factors that allow more complex control of the combustion process to optimize the combustion process. First of all, it is necessary to control the temperature in different places of the boiler. It is also necessary to control the end product of the combustion process. Then all three control possibilities will allow to optimize the combustion process in boilers of different power or heat generation devices. Therefore, there is a proposal to the authors to revise the experimental part of the paper taking into account this comment.
Response 1: Thank you for pointing this out. We appreciate the reviewer’s insightful comments, which have highlighted the importance of incorporating temperature distribution and combustion product analysis into the experimental design. The suggestion aligns with our goal of optimizing the combustion process through comprehensive control strategies. As emphasized, this study represents the first application of voiceprint technology to combustion monitoring and analysis. Our primary focus in this work was to establish a foundational framework for diagnosing combustion variations using acoustic signatures. This initial exploration aimed to validate the feasibility of leveraging acoustic features as a diagnostic tool. In response to the reviewer’s recommendation, we acknowledge the necessity of integrating temperature monitoring at multiple boiler locations and analysis of combustion end products in future experimental phases. These additions will enable a more thorough investigation of the combustion process, allowing for correlations between acoustic patterns, thermal dynamics, and chemical byproducts. Such an approach will enhance the robustness of our proposed optimization strategies and broaden their applicability to boilers of varying scales and configurations. The reviewer’s suggestion is well-received and will guide our subsequent research endeavors. We plan to address these aspects in detail in future studies, ensuring a holistic understanding of combustion processes and further refining the control methodologies. Once again, we thank the reviewer for their constructive feedback, which has significantly contributed to advancing the scope and impact of our work.
Comments 2: It should also be noted a large number of technical errors in the design of the article due to inattention of the authors. It is recommended to bring it in compliance with the rules of article design:
(1) line 35: the energy industry [1]. To adapt to the rapid growth and high variability of renewable en-
(2) line 36: ergy sources in the grid, as well as significant fluctuations in electricity demand [2], thermal
(3) line 39: operational status changes frequently, and faults such as Lean-blow out (LBO) [3-5], and ther-
(4) line 40: moacoustic instability [6-8] (TAI) occur frequently. These issues require higher standards for
(5) line 47: ation [9, 10], thermodynamic [11],and electrical detection technologies [12]. Among these, using the
(6) line 52: with the normal operation of the equipment [13, 14].
(7) line 56: such as LBO and TAI, has a long history and encompasses a wide range of directions. In
(8) line 58: ship between the flame acoustic spectra and burner diameters [15]; Yu measured the acoustic
(9) line 60: lyzing the changes in burning sound pressure level and spectral changes [16]; Singh found
(10) line 63: time-averaged and fluctuating temperatures [17]; Ahmed et al. analyzed and monitored the
(11) line 66: sations by analyzing the signals in both the time and frequency domains [18]; Mondal used
(12) line 69: flame, using this parameter to classify the flame into stable and unstable status [19]; Chan-
(13) line 72: thermoacoustic oscillations [13]. Past studies have primarily focused on analyzing the status
(14) line 84: recognition [20], electric motors [21], gas turbine combustor [22], bear [23], aero-engne [23], gasoline en-
(15) line 85: gines [25], and equipment status monitoring among others. In voiceprint monitoring, good
(16) line 88: frequency domain, Mel-Frequency Cepstral Coefficients [26] (MFCC), Gammatone Fre-
(17) line 93: recognition, MFCC features have demonstrated good recognition rates and robustness [27].
(18) line 114: since [28-30]. MFCC can simulate the nonlinear features of the human ear with a Mel filter
(19) line 115: bank and is usually used to extract signal features in sound signal processing [31]. It is the
(20) line 157: start of line without indentation
(21) line 163: where ΔMFCC is the first-order derivative of MFCC, N is the window size for the differ-
(22) line 175: start of line without indentation
(23) line 186: includes air tank (providing a stable oxygen supply), hydrogen tank (main test gas),
(24) line 187: methane tank (responsible for igniting hydrogen flame and avoiding tempering), steam
(25) line 188: generator (inhibiting hydrogen combustion rate) and its storage tank, electronic gas flow-
(26) line 196: per minute at standard conditions (0 °C, 1 atm). The third part is the combustion chamber,
(27) line 194: an accuracy of ±1%, and its range is 0-300 SLM; the accuracy of the device for measuring
(28) line 195: air is ±1%, and the range is 0-300SLM gas flowmeter. SLM is a unit used to express liters
(29) line 226: ard conditions (0 °C, 1 atm). Therefore, using the mass flow rate of air and fuel (respec-
(30) line 227: tively MA and MF) to calculate the value of the air coefficient,
(31) line 228: the fuel used in this study is 99% hydrogen and 99%
(32) line 272: study, has the following characteristics. As shown in Figure 3, when thermoacoustic os-
(33) line 299: such as Figs .5, the signal is a broadband signal with multiple dominant frequency distri-
(34) line 302: ing state, as shown in Figs. 6, and the main frequency becomes 221 Hz in the thermoacous-
(35) line 303: tic oscillation state, as shown in Figs. 7 bBy monitoring the change of its main frequency,
(36) line 312: Figure 6. Empering state frequency domain diagram.
(37) line 314: Figure 7. Hermoacoustic oscillation frequency domain diagram.
(38) line 328: In Figs. 8 ,Figs. 9, Figs. 10, the original data diagrams of thermoacoustic oscillation,
(39) line 356: vantages of model training are shown in Fig. 11 and Table 2.
(40) line 380: various states, and the diagnostic accuracy is as high as 93.49%. In contrast, although BP
(41) line 399: (1) The step index P shows high specificity in the tempering state, which can effec-
(42) line 402: safety. Through the monitoring of the frequency domain characteristics of the signal,
(43) line 404: (2) By monitoring the frequency domain characteristics of the signal, the change of
(44) line 408: sis.
(45) In Figures 8-10, the font size on the axes needs to be increased.
Response 2: Thank you for pointing this out. We sincerely apologize for the large number of technical and formatting errors in the original submission. Following the reviewer’s detailed and constructive feedback, we have carefully revised the manuscript and corrected all the issues accordingly:
(1)–(19): These issues primarily relate to reference formatting and citation inconsistencies. We have revised all relevant references to ensure proper placement, punctuation, and spacing between text and citations. All line breaks and sentence structures have also been checked to avoid any disruption caused by improperly placed references (Pages 1–4).
(20) & (22): Indentation has been added at the beginning of the corresponding paragraphs to meet the journal’s formatting requirements .
(21) & (23)–(31): Technical terms and symbols (SLM, MA, MF) have been clarified, formatted consistently, and units standardized throughout the manuscript
(32)–(44): Figure references and related sentence structures have been thoroughly revised to improve grammatical accuracy, clarity, and consistency of expression
(45): Axis label font sizes in Figures 8–10 have been increased to enhance readability, as per the reviewer’s suggestion.
We are truly grateful for the reviewer’s meticulous and insightful comments, which have significantly improved the technical quality and overall presentation of our manuscript.
2.Response to Comments on the Quality of English Language
Point 1: The English could be improved
Response 1: Thank you for your valuable suggestions regarding the quality of the English language. We have carefully reviewed the manuscript and addressed the issues related to awkward phrasing and unclear sentences. All identified problematic expressions have been revised to improve clarity, readability, and overall language quality throughout the manuscript.
- Additional clarifications
Thank you very much for your valuable comments. Regarding the technical errors mentioned in comment 2, we have carefully reviewed the manuscript and made the necessary changes to conform to the standard formatting style of the journal. In case the reviewer's comments were directed towards hyphenation issues in the text, we have re-proofread the original manuscript and found that no such issues existed in the original manuscript. However, we have ensured that the manuscript fully complies with the journal's formatting requirements to avoid any potential problems.
If further revisions are required due to my reading ability not understanding the true intent of the reviewers, please feel free to contact us and we will be happy to make any additional revisions. Once again, I thank the reviewers for their valuable comments on this paper.
Round 2
Reviewer 1 Report
Comments and Suggestions for Authors
Thank you for this revision. All of your edits in this version significantly improve the manuscript. However, errors are still present. The ones that stood out to me are:
- Line 136: the signal is segmented into "flames" - I believe you mean "frames".
- You refer to the "step index P" in many places, but in Line 104 you refer to the "step index Pi". You also refer to a "step data set Pi" in Figure 2. I think Line 104 is the only error here.
- In Line 247 you refer to the "ratio of oxygen to oxygen" - do you mean the ratio of fuel to oxygen?
- The x-axis of Figure 5 is labeled "frequency/s" and I think you mean "time/s"
- In Line 277, "Research" is misspelled.
- In Line 323, "Therefore, the state can be judged by frequency alone, but the specific state cannot be diagnosed." This sentence seems to contradict itself. I think you mean that the frequency domain signal can determine whether combustion is stable or unstable, but it cannot determine what kind of instability has occurred.
I think a thorough proofreading of the manuscript would improve it.
In addition to these minor issues, I have identified two major issues:
- As in the last version, there is not enough detail present for a reader to know how to replicate your experiment. You call the microphone a "shotgun microphone" but do not provide a brand or model number. It seems to me that the quality of your results could depend highly on the noise level, filtering characteristics, and directionality of the microphone. I gather that the microphone is placed near "the flame center" but I don't know where that is in terms of vertical distance above the burner face and radial distance away from the burner axis. Does the flame center position change between the different Table 1 conditions, and is the microphone moved to match?
- I am unclear how the "True Values" were determined for the confusion matrices for the neural networks in Section 4. Is the true state of the system determined from the step index P? Or through visual assessment of the camera images? You mention combustion state images, but do not state how they were used in this study. Providing image examples for different states in a figure could be useful.
Author Response
Comments 1: Thank you for this revision. All of your edits in this version significantly improve the manuscript. However, errors are still present. The ones that stood out to me are:
- Line 136: the signal is segmented into "flames" - I believe you mean "frames".
- You refer to the "step index P" in many places, but in Line 104 you refer to the "step index Pi". You also refer to a "step data set Pi" in Figure 2. I think Line 104 is the only error here.
- In Line 247 you refer to the "ratio of oxygen to oxygen" - do you mean the ratio of fuel to oxygen?
- The x-axis of Figure 5 is labeled "frequency/s" and I think you mean "time/s"
- In Line 277, "Research" is misspelled.
- In Line 323, "Therefore, the state can be judged by frequency alone, but the specific state cannot be diagnosed." This sentence seems to contradict itself. I think you mean that the frequency domain signal can determine whether combustion is stable or unstable, but it cannot determine what kind of instability has occurred.
Response 1: Thank you for your suggestions. According to your suggestions for change, I modified some paragraphs. First, on line 136, I mistakenly wrote the frame as flame and changed it. Second, the 104 row step data set you mentioned can better express the meaning of the study, which has been changed. Third, in line 247, the meaning of the research comparison is to keep the air coefficient unchanged and change the steam flow in the pipeline. In the fourth Figure 5, the x-axis should be frequency/hz, which has been changed. Fifthly, 277 lines of spelling errors have been corrected. Finally, the description you mentioned can more accurately express the conclusion of this study, and the corresponding paragraphs have been modified.
Comments 2:As in the last version, there is not enough detail present for a reader to know how to replicate your experiment. You call the microphone a "shotgun microphone" but do not provide a brand or model number. It seems to me that the quality of your results could depend highly on the noise level, filtering characteristics, and directionality of the microphone. I gather that the microphone is placed near "the flame center" but I don't know where that is in terms of vertical distance above the burner face and radial distance away from the burner axis. Does the flame center position change between the different Table 1 conditions, and is the microphone moved to match?
Response 2: Thank you for your suggestion. Based on your suggestion, I have included a specific description of the microphone in this research paper, which uses a Rodes microphone, which has a better effect on the reception and suppression of external noise, and a further detailed description of the location of the microphone.
Comments 3:I am unclear how the "True Values" were determined for the confusion matrices for the neural networks in Section 4. Is the true state of the system determined from the step index P? Or through visual assessment of the camera images? You mention combustion state images, but do not state how they were used in this study. Providing image examples for different states in a figure could be useful.
Response 3: Thank you for your suggestion. According to your suggestion, I added a specific explanation of the real value of model training in Section 4 results and discussion. The label of the true value is marked manually to judge the accuracy of the state, in which the step index set Pi is used as the diagnosis basis of the state change, and the model is not trained. At the same time, the burner used in this study is a confidential project, and the specific image cannot be provided for the time being, and the description of the combustion image part has been changed. Thank you for your review and understanding.
Reviewer 2 Report
Comments and Suggestions for Authors
The authors made a number of changes to the text, which improved the quality of the article. However, a number of technical errors remain to be corrected.
The text on lines 132, 165, 179, 187 should start without indentation. Also, commas should be placed after a number of expressions, where there is an explanation of symbols in the following lines.
After the technical remarks are corrected, the article can be published.
Author Response
Comments 1: The authors made a number of changes to the text, which improved the quality of the article. However, a number of technical errors remain to be corrected.
The text on lines 132, 165, 179, 187 should start without indentation. Also, commas should be placed after a number of expressions, where there is an explanation of symbols in the following lines.
Response 1:Thanks for the comments from the review experts. According to the comments of the review experts, I have indented the corresponding questions in lines 132, 165, 179 and 187. At the same time, the nonstandard use of commas in this paper is further revised and modified.